# LOCAL UNCERTAINTY SMOOTHING FOR FEW-SHOT OOD DETECTION

## ABSTRACT

Few-shot out-of-distribution (OOD) detection has become a critical research direction for the practical deployment of machine learning systems. Existing approaches commonly rely on auxiliary outlier data derived from in-distribution (ID) samples, such as using local image patches from training data to simulate OOD features. However, these artificially constructed OOD samples differ substantially from real OOD instances, leading to unstable learning when trained with hard OOD labels. To address this challenge, we propose a Local Uncertainty Smoothing (LUS) framework for few-shot OOD detection. Our method incorporates label smoothing and local uncertainty measure to facilitate a smooth transition between the reference distribution of local image categories, based on a general knowledge model and the target OOD distribution. This approach ensures strong OOD detection performance while preserving the model's ability to capture detailed local-level semantic features. Furthermore, we theoretically analyze the relevance of local uncertainty from the perspective of a generalization error bound (GEB). This reveals a concrete relationship between our local uncertainty measure and the KL divergence observed during training. Accordingly, we propose a patch-wise local uncertainty to effectively identify suitable soft labels for the model throughout the learning process, achieving superior OOD detection performance. Extensive experiments on real-world OOD benchmarks validate the effectiveness of our approach. Code will be made publicly available.

## 1 INTRODUCTION

Deep learning systems are primarily built upon the theoretical framework of the independent and identically distributed assumption, which presumes identical probability distributions between training and test data. However, real-world data acquisition systems inevitably face challenges of distributional shifts, where such discrepancies in probability distributions may pose significant safety risks, particularly in safety-critical applications such as autonomous driving and medical diagnosis. In response to these challenges, diverse methodologies for OOD evaluation have proliferated. Notably, with the advent of prompt learning in pre-trained vision-language models (Radford et al., 2021), CLIP-based prompt tuning (Zhou et al., 2022b;a) has been strategically adapted for OOD detection tasks, catalyzing growing research interest in leveraging prompt learning paradigms for enhanced OOD detection capabilities.

Recent advances have leveraged auxiliary OOD datasets to improve OOD detection. For instance, as shown in our Fig. 1a. (Hendrycks et al., 2018) demonstrated that assigning one-hot labels to entire ID images and uniform label distributions to entire OOD images can lead to effective OOD detection. Other methods (Bai et al., 2024), (Miyai et al., 2024) have also made progress by generating OOD data using only ID data during training. However, we reveal a critical limitation of a learning paradigms that treat local patches as OOD samples and assign them uniform label distributions - this labeling strategy is inherently unsuitable. Specifically, the representation of finer-grained features in OOD local data presents significant challenges and often results in classification errors. Using uniform distribution for local OOD features in such a setting will negatively impact ID classification and OOD detection performance. For example, as shown in our Fig. 1b, our results indicate that background images of lions show significant correlation with features of cliffs and stone walls. Using such ID-like images as OOD features adversely affects the model in both ID classification and OOD detection. It impairs the model's ability to recognize features of cliffs and stone walls

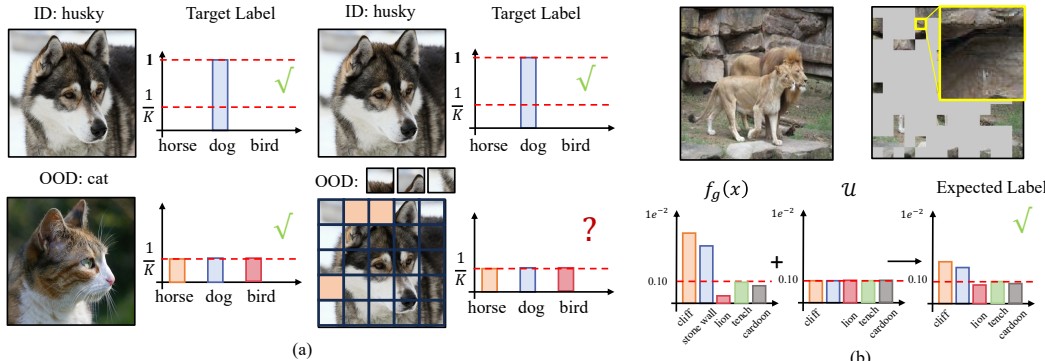

Figure 1: Problematic Label Assignment. (a) illustrates the OOD labeling issue arising from the use of an auxiliary training set and local OOD features sampled from ID data, highlighting the need to reconsider the labeling strategy for such features. (b) depicts a typical case of this local OOD feature labeling problem and presents the underlying rationale for a revised labeling approach.

during ID classification, leading to a decline in ID classification accuracy. Meanwhile, it blurs the feature distribution distinction between ID and OOD data, thereby compromising OOD detection performance. To overcome this limitation, it is imperative to preserve partial semantic features from the local OOD images. As demonstrated in Fig. 1b, we contend that the authentic distribution of patch-level OOD data is a label distribution that reduces confidence in general categorical knowledge, as opposed to a simple uniform distribution.

In this study, we propose the Local Uncertainty Smoothing framework. To preserve the ability of patch-level data to retain semantic understanding across different labels, we introduce a general category knowledge prior to serve as a reference distribution. Subsequently, we construct soft labels by incorporating label smoothing and a novel patch-wise local uncertainty mechanism. These soft labels are designed to simultaneously maintain the sensitivity of patch-level data to semantic distinctions while enhance OOD detection capability. Furthermore, inspired by generalization error bound theory, we investigate the relationship between patch-wise local uncertainty and KL divergence during training. This theoretical foundation enables our model to adaptively determine optimal soft label assignments for each OOD patch. Extensive experiments demonstrate that the proposed framework yields superior OOD detection performance.

- We propose an intuitive and novel OOD soft label construction paradigm for few-shot OOD detection. Based on the label smoothing, we derive a Local Uncertainty Smoothing (`LUS`) framework to assign reasonable OOD labels for the ID local patches. This offers a new understand of the ood local features and effectively improve the ood detection perfermance.

- We propose a patch-wise local uncertainty metric based on the covariance between the uncertainty and the KL divergence observed during training. This offers a theoretical guarantee for the relevance of local uncertainty from the perspective of the generalization error upper bound.

- We develop a novel dynamic iterative learning methodology which refines the uncertainty metric to progressively learn superior soft labels. Extensive experimental results validate our theoretical analysis and demonstrate the superior performance of the proposed approach.

## 2 RELATED WORK

**OOD Detection with Pre-trained Vision-language Models.** OOD detection aims to identify inputs from unknown classes absent during training, ensuring model reliability. Traditional methods leverage confidence scores like MSP (Hendrycks & Gimpel, 2016), perturbation-enhanced ODIN (Liang et al., 2017), or feature-space metrics such as Mahalanobis distance (Lee et al., 2018). Recent advances exploit vision-language models (VLMs) like CLIP, which align visual and textual embeddings for zero-shot inference. Zero-shot CLIP-based approaches (Esmaeilpour et al., 2022)utilize pre-trained prompts to estimate OOD score with temperature-scaled softmax to enhance separability without fine-tuning. Beyond zero-shot, fine-tuned methods (Du et al., 2022), (Tao et al., 2023) incorporate ID data

for task-specific calibration, albeit with increased computational costs, such as CLIPN (Wang et al., 2023) refine detection via negative prompt generation. Recently, a promising direction is few-shot OOD detection, which balances efficiency and performance by leveraging minimal in-distribution samples. Currently, the mainstream approaches leveraging CLIP-based prompt learning for OOD few-shot detection primarily follow two methodologies. One is the LoCoOP (Miyai et al., 2024) method, which enforces entropy uniformity for distribution alignment, and the other adopts a K+1 class formulation that introduces an auxiliary dimension to learn negative prompts. Variants of the latter include techniques such as id-like (Bai et al., 2024), which reduces the number of negative prompts by learning common features across categories, and the NegPrompt (Li et al., 2024) method, which employs shared class-specific contexts for both positive and negative prompt construction.

**Prior knowledge transfer.** Prior knowledge transfer has been extensively utilized across various domains. In the context of CLIP, prior knowledge transfer has been widely adopted to address catastrophic forgetting in tasks such as few-shot accuracy prediction and domain generalization. For instance, ProGrad (Zhu et al., 2023) ensures alignment between the learning direction of trainable task-specific knowledge and general knowledge (hand-crafted prompts) during prompt tuning, thereby preserving existing knowledge while acquiring new capabilities. Similarly, while ProGrad discards conflicting updates by optimizing prompts toward aligned directions, KgCoOp (Yao et al., 2023) avoids knowledge discardment by introducing a Euclidean distance-based loss to constrain trainable task-specific knowledge to remain proximal to general knowledge. Inspired by these approaches, our work investigates catastrophic forgetting in OOD detection and explores how to effectively leverage prior knowledge transfer to enhance OOD detection performance.

## 3 PRELIMINARIES

We partition the dataset into a training set $\mathcal{D}_{\text{train}} = (\mathcal{D}_{\text{train}}^{\text{ID}}, \mathcal{D}_{\text{train}}^{\text{OOD}})$ and a validation set $\mathcal{D}_{\text{test}} = (\mathcal{D}_{\text{test}}^{\text{ID}}, \mathcal{D}_{\text{test}}^{\text{OOD}})$, where the ID components $\mathcal{D}_{\text{train}}^{\text{ID}}$ and $\mathcal{D}_{\text{test}}^{\text{ID}}$ adhere to the joint data-label distribution $(x_i, y_i)$ with explicit sample-label pairs $(x, y)$, while the OOD components $\mathcal{D}_{\text{train}}^{\text{OOD}}$ and $\mathcal{D}_{\text{test}}^{\text{OOD}}$ are sampled from unknown $P_{\text{OOD}}$. But current few-shot learning paradigms increasingly avoid reliance on external OOD datasets. Methods exemplified by LoCoOP (Miyai et al., 2024) leverage CLIP's inherent prior knowledge to synthesize OOD samples from ID data through patch-based strategies, while approaches like id-like (Bai et al., 2024) generate OOD representations via random cropping of ID samples. Consequently, the majority of OOD data in few-shot scenarios originates from systematic transformations of ID data rather than external collections.

**Zero-Shot OOD Detection.** Given a pre-trained vision-language model (Radford et al., 2021) with image encoder $\phi_I(\cdot)$ and text encoder $\phi_T(\cdot)$. The MCM (Ming et al., 2022) method computes OOD scores through cross-modal alignment. MCM's zero-shot capability stems from leveraging the pre-trained cross-modal alignment without fine-tuning on ID data. The key hypothesis is that OOD samples exhibit lower maximum similarity due to semantic misalignment with ID class prompts. For an input image $\mathbf{x}$, the scoring function $S(\mathbf{x})$ is defined as:

$$S(\mathbf{x}) = \max_i \frac{\exp\left(\langle \phi_I(\mathbf{x}), \phi_T(\mathbf{t}_i) \rangle / \tau\right)}{\sum_{j=1}^{C} \exp\left(\langle \phi_I(\mathbf{x}), \phi_T(\mathbf{t}_j) \rangle / \tau\right)} \tag{1}$$

where $\mathbf{t}_i$ represents the hand-crafted prompt for class $i$, and $\tau$ is the temperature parameter to be set as 1. The OOD decision rule follows:

$$\mathcal{F}(\mathbf{x}) = \begin{cases} \text{ID}, & S(\mathbf{x}) \geq \tau \\ \text{OOD}, & S(\mathbf{x}) < \tau \end{cases} \tag{2}$$

**Prompt learning for OOD detection.** In contrast to conventional prompt learning frameworks like CoOP (Zhou et al., 2022b), our method adopts the state-of-the-art OOD detection approach LoCoOp (Miyai et al., 2024) . This framework operates without introducing auxiliary dimensions, instead directly fine-tuning the original classification logit distribution. To address the challenge of detecting real OOD samples under distributional uncertainty, we construct auxiliary OOD data by leveraging low-similarity patches from ID samples under the LoCoOp (Miyai et al., 2024) paradigm. A uniform label distribution $\mathcal{U}$ is imposed to suppress the original distribution of OOD data. Our final loss

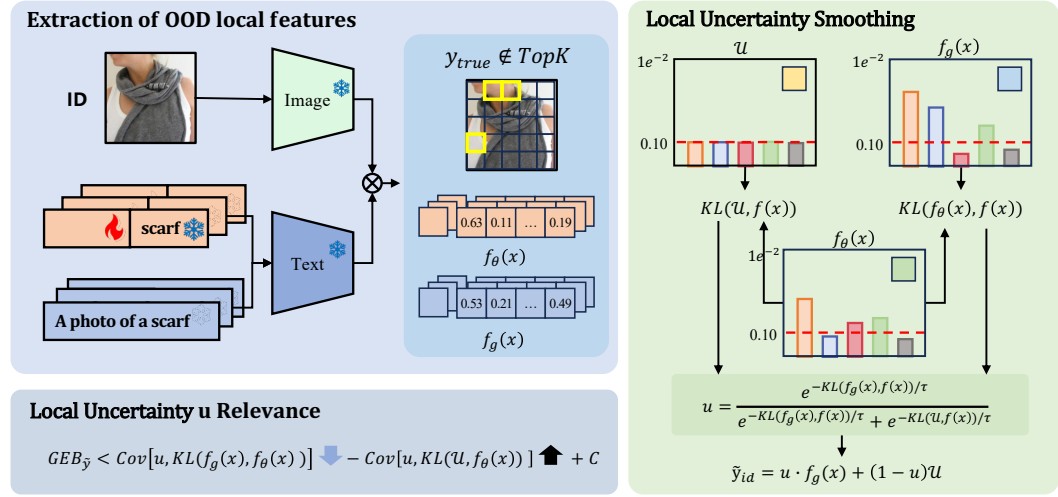

Figure 2: Overview of our framework. Our method begins by following LoCoOp for the extraction of OOD local features. Then, we introduce a Local Uncertainty Smoothing approach to reformulate the OOD soft labels. Subsequently, we theoretically explore the relevance of the local uncertainty and propose a local uncertainty mechanism that enables the model to adaptively identify optimal soft labels.

function is formulated as:

$$
\begin{aligned}
\mathcal{L}_{train} &= \mathcal{L}_{\text{CE}} + \lambda \mathcal{L}_{\text{OOD}} \\
&= \mathbb{E}_{\mathcal{D}_{\text{train}}^{\text{ID}}}\left[ -y_{true} \log f_\theta(x) \right] + \alpha \mathbb{E}_{\mathcal{D}_{\text{train}}^{\text{OOD}}}\left[ -H(f_\theta(x)) \right] \\
&= \mathbb{E}_{\mathcal{D}_{\text{train}}^{\text{ID}}} KL\left( y, f_\theta(x) \right) + \alpha \mathbb{E}_{\mathcal{D}_{\text{train}}^{\text{OOD}}} KL\left( \mathcal{U}, f_\theta(x) \right) + \alpha \log K
\end{aligned}
\tag{3}
$$

Here $f_\theta(x)$ denotes the training model, $\alpha$ controls the OOD loss strength, and $K$ is the number of ID classes.

**Extraction of OOD local features.** To identify regions that are irrelevant to the ID categories, we perform a selection from the complete index set $I = \{0, 1, 2, \ldots, H \times W - 1\}$, with $H$ and $W$ indicating the spatial dimensions of the feature map. Throughout training, the classification probability for each region $i$ is determined by measuring the similarity between its visual feature $f_\theta^{(i)}$ and the textual embeddings of the ID classes. A region is considered ID-irrelevant and included in the set $R$ if its ground-truth category does not appear within the top-$K$ classes with the highest predicted probabilities. This is formally expressed as:

$$
R = \left\{ i \in I : \text{rank}(\pi^{(i)}(y \mid x; \boldsymbol{\omega})) > K \right\}, \quad (6)
$$

Here, $\pi^{(i)}(y \mid x; \boldsymbol{\omega})$ signifies the predicted probability assigned to the true label for the $i$-th region, and $\text{rank}(\pi^{(i)}(y \mid x; \boldsymbol{\omega}))$ indicates the ordinal position of that true label when all ID class probabilities are sorted in descending order.

## 4 METHOD

In the previous section, we discussed the limitations of our baseline approach, which utilized entire images to learn from one-shot distributions while training on patch-level OOD data under a uniform distribution. This misalignment compromises the model's generalization capability. To overcome this issue, we propose local uncertainty smoothing method which refining the label distribution for

OOD patches by incorporating reference distribution of local image categories, based on a general knowledge $f_g$, rather than enforcing a uniform distribution indiscriminately. To further optimize the patch-level OOD labels, we introduce Patch-wise local Uncertainty. This method dynamically smooths the label distribution between reference distribution and uniform distribution by integrating local uncertainty. Our design of this uncertainty-aware mechanism is motivated by theoretical insights into generalization error bounds, which reveal a critical relationship between uncertainty and KL divergence observed during training. Our framework is shown in Fig. 2.

## 4.1 Local Uncertainty Smoothing.

Building upon the label smoothing formulation, we aim to smooth the patch-level reference distribution gradually toward a uniform distribution over the OOD detection, while retaining the model's inherent patch-level semantic knowledge. Accordingly, we define the soft label for OOD data as follows:

$$\widetilde{y} = u \cdot f_g(x) + (1 - u) \cdot \mathcal{U} \tag{4}$$

where $u$ denotes an uncertainty method, $\mathcal{U}$ is a uniform distribution and $f_g(x)$ represents a general knowledge model that denotes the semantic understanding of patch-level OOD data. When the value of $u$ is relatively small, the label assignment treats the current patch as a typical OOD example with minimal semantic relevance to any known class, thereby effectively representing a feature devoid of categorical associations. Conversely, a larger $u$ indicates that the patch is perceived as highly relevant to a specific class label.

Based on the definition of patch-level OOD soft labels, we construct the corresponding OOD loss function using the KL divergence and the formula is as follows:

$$\mathcal{L}_{\text{OOD}} = \mathbb{E}_{\mathcal{D}_{\text{train}}^{\text{OOD}}} \left[ \ell(\widetilde{y}, f_\theta(x)) \right] \tag{5}$$

Following the setup in LoCoOp, we use the same ID loss for the entire image. The overall objective function with full images as ID data and patch-level regions as OOD data is defined as follows:

$$\mathcal{L} = \mathcal{L}_{\text{ID}} + \lambda \mathcal{L}_{\text{OOD}} \tag{6}$$

where $\mathcal{L}_{\text{ID}}$ is as same as the $\mathcal{L}_{\text{CE}}$ and $\lambda$ is a hyperparameter.

## 4.2 Patch-wise Local Uncertainty

We have redefined the data labels at the patch level. However, quantifying the relevance between $f_g(x)$ and $\mathcal{U}$ remains challenging, and the question of how to determine an optimal $u$ for OOD detection merits further investigation. Therefore, determining an appropriate value of $u$ is critical after incorporating the label smoothing strategy described above. In contrast to traditional approaches, we propose to examine the relationship between $u$ and the model through the lens of generalization error minimization. This perspective allows us to develop a dynamic weighting scheme that enhances OOD detection performance.

In machine learning, the concept of the generalization error bound describes a theoretical limit on model performance when applied to unseen data (Niyogi & Girosi, 1996). A tighter bound generally indicates better expected performance on data from unknown distributions. In this section, the generalization error of an OOD detector $f$ can be defined as follows:

$$\text{GError}(f, \widetilde{y}^*) = \mathbb{E}_{x \sim \mathcal{D}}[\ell(\widetilde{y}^*, f_\theta(x))]. \tag{7}$$

It should be emphasized that our analysis of generalization error bounds pertains to OOD data that follows the same distribution as the patch-level OOD regions, under the assumption that an optimal parameter $u^*$ exists to minimize these bounds. And and $\widetilde{y}^*$ denotes the optimal soft label under the optimal $u^*$. Building on this foundation, we formalize our theoretical framework.

**Theorem 1.** *(Dynamic Smoothness Uncertainty Relevance). For any hypothesis $f \in \mathcal{F}$ and $\mathcal{F}$ denotes the optimisation space for prompt learning, given a data point x, we possess an optimal soft label $\widetilde{y}^*$ under an optimal $u^*$. We holds with a generalization error upper bound:*

Table 1: Benchmark OOD detection performance on ImageNet-1K as the ID dataset across CLIP-based architectures. Results are reported as mean across three randomized seeds. ViT-B/16 is adopted as the reference image encoder.

| Method | Backbone | iNaturalist | | SUN | | OOD Dataset Places | | Texture | | Average | |
|---|---|---|---|---|---|---|---|---|---|---|---|
| | | FPR95 | AUROC | FPR95 | AUROC | FPR95 | AUROC | FPR95 | AUROC | FPR95 | AUROC |
| | | | | | | Full/Sub Data Fine-tune | | | | | |
| MSP | CLIP-B/16 | 40.89 | 88.63 | 65.81 | 81.24 | 67.90 | 80.14 | 64.96 | 78.16 | 57.92 | 82.31 |
| Energy | CLIP-B/16 | 29.75 | 94.68 | 34.28 | 93.15 | 56.40 | 85.60 | 51.35 | 88.00 | 45.83 | 89.43 |
| ODIN | CLIP-B/16 | 30.22 | 94.65 | 54.04 | 87.17 | 55.06 | 85.54 | 51.67 | 87.85 | 45.65 | 89.35 |
| Fort/MSP | CLIP-B/16 | 54.05 | 87.43 | 54.12 | 86.37 | 72.98 | 78.03 | 68.85 | 79.06 | 65.29 | 81.51 |
| VOS | CLIP-B/16 | 31.65 | 94.53 | 43.03 | 91.92 | 41.62 | 90.23 | 56.67 | 86.74 | 43.31 | 90.50 |
| NPOS | CLIP-B/16 | 16.58 | 96.19 | 43.77 | 90.44 | 45.27 | 89.44 | 46.12 | 88.80 | 35.99 | 91.48 |
| CLIPN | CLIP-B/16 | 23.94 | 95.27 | 26.17 | 93.93 | 33.45 | 92.28 | 40.83 | 90.93 | 32.74 | 92.83 |
| | | | | | | Zero-shot | | | | | |
| MCM | CLIP-B/16 | 30.91 | 94.61 | 37.67 | 92.56 | 44.69 | 89.77 | 57.77 | 86.11 | 44.46 | 90.16 |
| | | | | | | One-shot | | | | | |
| CoOp | CLIP-B/16 | 43.38 | 91.26 | 38.53 | 91.95 | 46.68 | 89.09 | 50.64 | 87.83 | 46.90 | 89.39 |
| id-like | CLIP-B/16 | **12.07** | **97.65** | 40.55 | 91.07 | 47.94 | 88.31 | 38.34 | 89.67 | 34.72 | 91.67 |
| NegPrompt | CLIP-B/16 | 65.03 | 84.56 | 44.39 | 89.63 | 51.31 | 86.55 | 67.60 | 63.76 | 62.08 | 81.13 |
| LoCoOp$_{MCM}$ | CLIP-B/16 | 32.05 | 93.61 | 33.60 | 93.01 | 41.29 | 90.05 | 51.51 | 88.62 | 39.61 | 91.32 |
| LoCoOp$_{GL}$ | CLIP-B/16 | 19.67 | 95.83 | 25.73 | 94.00 | 34.95 | 91.06 | 52.73 | 87.03 | 33.27 | 91.98 |
| LUS$_{MCM}$ | CLIP-B/16 | 29.54 | 94.34 | **28.73** | **94.13** | **35.09** | **91.48** | 49.70 | 88.86 | 35.76 | 92.20 |
| LUS$_{GL}$ | CLIP-B/16 | 18.32 | 96.31 | **25.27** | **94.70** | **34.02** | **91.65** | 51.10 | 86.99 | **32.18** | **92.41** |
| | | | | | | 16-shot | | | | | |
| CoOp | CLIP-B/16 | 35.36 | 92.60 | 37.06 | 92.27 | 45.38 | 89.15 | 43.74 | 89.68 | 41.49 | 90.48 |
| id-like | CLIP-B/16 | 13.94 | 95.42 | 42.28 | 89.42 | 53.25 | 85.44 | 18.16 | 93.78 | 31.91 | 91.01 |
| NegPrompt | CLIP-B/16 | 37.79 | 90.49 | 32.11 | 92.25 | 35.52 | 91.16 | 43.93 | 88.38 | 37.34 | 90.57 |
| LoCoOp$_{MCM}$ | CLIP-B/16 | 24.38 | 94.86 | 30.85 | 93.68 | 37.45 | 91.24 | 43.42 | 90.28 | 34.03 | 92.51 |
| LoCoOp$_{GL}$ | CLIP-B/16 | 13.99 | 96.83 | 23.37 | 94.78 | 31.87 | 91.87 | 45.14 | 88.18 | 28.59 | 92.92 |
| LUS$_{MCM}$ | CLIP-B/16 | 25.08 | 95.02 | **30.16** | **93.75** | **37.05** | **91.28** | 41.61 | 90.87 | 33.48 | 92.73 |
| LUS$_{GL}$ | CLIP-B/16 | 15.87 | 96.61 | **21.92** | **94.97** | **30.77** | **92.14** | 42.54 | 89.51 | **27.77** | **93.31** |

$$GError(f, \widetilde{y}^*)) \leq Cov\left[u^*, KL(f_g(x), f_\theta(x))\right] - Cov\left[u^*, KL(\mathcal{U}, f_\theta(x))\right] + C \quad (8)$$

*where $Cov[u^*, KL(f_g(x), f_\theta(x))]$ and $Cov[u^*, KL(\mathcal{U}, f_\theta(x))]$ is the covariance between Dynamic weight $u^*$, loss function of $KL(\mathcal{U}, f_\theta(x))$ and loss function of $KL(\mathcal{U}, f_\theta(x))$. C is a term independent of $u^*$.*

A detailed proof is provided in Section Appendix A. It should be emphasized that our method focuses on the correlation between the optimal $u^*$ and the model. In the context of label smoothing, the $C$ term has been extensively studied and can be approximated as a constant (Yuan et al., 2020). Moreover, $C$ is formally independent of the optimal $u^*$. Therefore, the theoretical analysis in this section is confined to the correlation concerning $u^*$, while a more detailed discussion of the $C$ term is provided later in the discussion section.

**Remark.** In our framework, the KL divergence is adopted as the loss function. Within our theoretical setup and under the stated assumptions, reducing the generalization error bound requires satisfying the conditions that $Cov[u^*, KL(f_g(x), f_\theta(x))] < 0$ and $Cov[u^*, KL(\mathcal{U}, f_\theta(x))] > 0$. This leads to two corollaries for the design of u that it must be negatively correlated with the $KL(f_g(x), f_\theta(x))$, and positively correlated with the $KL(\mathcal{U}, f_\theta(x))$.

Based on the theoretical foundation established above, we define the label weights $u$ for our model. Prior to each iteration of the loss computation, we calculate a preliminary $KL(f_g(x), f_\theta(x))$ and $KL(\mathcal{U}, f_\theta(x))$, which informs the following formulation of the dynamic smoothness uncertainty:

$$u = \frac{e^{-KL(f_g(x), f_\theta(x))}/\tau}{e^{-KL(f_g(x), f_\theta(x))/\tau} + e^{-KL(\mathcal{U}, f_\theta(x))/\tau}} \quad (9)$$

where $\tau$ is a hyperparameter. $\tau$ reflects the sensitivity of the label weighting to the loss function. We performed detailed experimental validation of this behavior in Section 3.

Table 2: Accuracy comparison of ID on the ImageNet-1K validation data for few-shot object detection.

| Shot | Method | Accuracy (%) |
|---|---|---|
| 1-Shot | CoOp | 66.23 |
| | LoCoOp | 68.10 |
| | Ours | 68.70 |
| 16-Shot | CoOp | 72.10 |
| | LoCoOp | 71.10 |
| | Ours | 71.40 |

Table 3: Performance comparison with LoCoOp on hard ood dataset. Our first row represents the id dataset and the second row represents the ood dataset.

| Method | ImageNet10 ImageNet20 | | ImageNet20 ImageNet10 | |
|---|---|---|---|---|
| | FPR95 | AUROC | FPR95 | AUROC |
| LoCoOp | 28.20 | 92.75 | 34.40 | 92.34 |
| Ours | 5.70 | 98.60 | 16.10 | 97.66 |

After defining $u$, and given the continuous iterative nature of model training, the optimal value of $u$ typically varies over time with respect to the evolving $f_\theta(x)$. To account for this, we design $u$ as a time-dependent uncertainty measure that is recomputed after each backward propagation of the loss. And dynamic smoothness uncertainty is as follows:

$$u^t = \frac{e^{-KL(f_g(x), f_\theta^t(x))}/\tau}{e^{-KL(f_g(x), f_\theta^t(x))/\tau} + e^{-KL(\mathcal{U}, f_\theta^t(x))/\tau}} \qquad (10)$$

where $u^t$ recalculate based on $f^t$ after each backpropagation of the gradient. And $f_\theta^t(x)$ represents the training model after t iterations of gradient backpropagation. Under this configuration, $u$ dynamically adapts to the current $f_\theta(x)$, thereby facilitating the search for optimal soft labels. As a result, the joint optimization of $u$ and the learning process of $f_\theta(x)$ mutually reinforce each other, ultimately converging to both suitable soft labels and a well-trained model.

## 5 EXPERIMENTS

### 5.1 EXPERIMENTAL SETUP

**Datasets.** We evaluate our method across three distinct OOD detection benchmarks to comprehensively assess performance under varying scenarios. First, following standard protocols, we employ ImageNet-1K (Deng et al., 2009) as the ID dataset and test on widely-used OOD benchmarks including iNaturalist (Van Horn et al., 2018), SUN (Xiao et al., 2010), Places (Zhou et al., 2018), and Textures (Cimpoi et al., 2014) with few-shot training. Second, to rigorously examine hard OOD detection, we adopt the MCM (Esmaeilpour et al., 2022) ImageNet-10 and ImageNet-20 setup, where ImageNet-10 mimics CIFAR-10's class distribution with high-resolution images, and ImageNet-20 introduces semantically similar near-OOD classes. More experiments set can be found in the Appendix C.

**Implementation details.** Our implementation adheres to the LoCoOp framework with CLIP-ViT-B/16 (Dosovitskiy et al., 2020) as the backbone, where the feature maps exhibit a spatial resolution of 16×16. The key hyperparameters are empirically configured as follows: the neighborhood size K=200 across all experiments, and the regularization weight $\lambda$ =0.5. Additional training specifications include 50 epochs with a base learning rate of 0.002, batch size of 32, and prompt token length N=16. All experiments are conducted on a single NVIDIA A6000 GPU to ensure hardware consistency.

**Baselines and Evaluation.** Our comparative analysis encompasses three methodological paradigms. Fully-supervised approaches such as MSP (Hendrycks & Gimpel, 2016), Fort/MSP (Fort et al., 2021), Energy (Liu et al., 2020), ODIN (Liang et al., 2017), VOS (Du et al., 2022), and NPOS (Tao et al., 2023), zero-shot approaches represented by MCM (Ming et al., 2022), and few-shot methods including CoOp (Zhou et al., 2022b) and LoCoOp. All methods employ the CLIP ViT-B-16 backbone to ensure equitable comparison. Performance evaluation leverages three standard metrics FPR95, AUROC, and ID classification accuracy which enable comprehensive assessment of detection capabilities.

Table 4: Ablation analysis of local uncertainty $u$ with other different uncertainty method on OOD datasets with average results.

| Method | AVG | |
|---|---|---|
| | FPR95 | AUROC |
| Static weight | 40.54 | 91.62 |
| Entropy | 42.91 | 90.94 |
| MaxLogit | 38.42 | 91.63 |
| Ours | 35.76 | 92.20 |

Table 5: Ablation analysis of framework components with $f_g(x)$ and $u$ on OOD datasets with average results.

| $f_g(x)$ | u | AVG | |
|---|---|---|---|
| | | FPR95 | AUROC |
| ✗ | ✗ | 39.61 | 91.32 |
| ✓ | ✗ | 39.54 | 91.50 |
| ✓ | ✓ | 35.76 | 92.20 |

## 5.2 MAIN RESULTS

**ImageNet-1k as ID dataset.** Table 1 summarizes our OOD detection performance using ImageNet-1K as ID data. Our proposed framework, which design soft labels suitable for patch-level OOD data, achieves state-of-the-art performance across both 1-shot and 16-shot configurations. Our approach achieves comprehensive improvements, whether for ID classification or OOD detection. The ID classification result as shown in Table 2. And demonstrates significant improvements with average FPR95 and AUROC scores of 35.76 and 92.20 in 1-shot settings. This method outperforming conventional OOD detection methods and even surpassing the original zero-shot CLIP baseline and LoCoOp. Moreover, on several other datasets set up in Openood using ImageNet-1k as id, our method also achieve superior performance compared to our baseline. More our experimental results are presented in the Appendix E.

**Comparisons on hard OOD detection.** Our method achieves robust performance on small hard-OOD datasets while maintaining the ability to recognize patch-level feature information. As shown in Table 3, our approach consistently outperforms the baseline and significantly surpasses LoCoOp, which exhibits considerable performance degradation on these datasets. These results confirm that our framework effectively preserves the discriminative power of categorical features rather than simply enforcing alignment toward a uniform distribution.

## 5.3 ABLATION STUDY

**Impact of Components of Local Uncertainty Smoothing.** We performed a comprehensive evaluation of the effectiveness of our label smoothing strategy. The results, presented in Table 5, show a significant improvement over the baseline without label smoothing. Notably, even with static weighting and without dynamic smoothing uncertainty, our label smoothing approach enhances object detection performance. Moreover, the introduction of dynamic smoothing uncertainty further optimizes the results, achieving the best overall performance and confirming the efficacy of the proposed method. Based on this work, we include in the Appendix D the selection and experimental results of various $f_g$. Our results demonstrate that a well-chosen general knowledge model—one with strong prior understanding of the dataset—can lead to greater performance gains in OOD detection.

**Impact of Local Uncertainty Method.** We conducted targeted experiments to evaluate the efficacy of the dynamic uncertainty smoothing component within our proposed method. The results are summarized in Table 4. Without dynamic weighting, even a simple equal weight strategy already yielded a marked improvement in OOD detection performance compared to static weighting approaches. Furthermore, we compared our dynamic uncertainty smoothing mechanism against other uncertainty-based fusion methods, including those based on entropy and maximum logit to further verify its advantage. Detailed configurations of these baseline methods are provided in the Appendix F. Our results confirm the effectiveness of dynamically designed weighting schemes derived from correlations in smoothed uncertainty.

**Impact of Temperature coefficient $\tau$.** We present the results for the parameter $\tau$ evaluated over a range of values (0, 0.1, 0.2, 0.4, 0.8, 1, 2, 4, 8, 10) in Fig. 3. The parameter $\tau$ controls the sensitivity balance between the two KL divergence terms. The results show that a small $\tau$ amplifies the KL divergence values, making their relative ratio more prominent. However, this configuration yields suboptimal performance. We analyze that this is due to the iterative nature of the model training, there is an excessively large ratio may lead to instability in the uncertainty measure $u$, which can cause erratic changes in the soft labels and hinder stable model learning. Conversely, an overly small

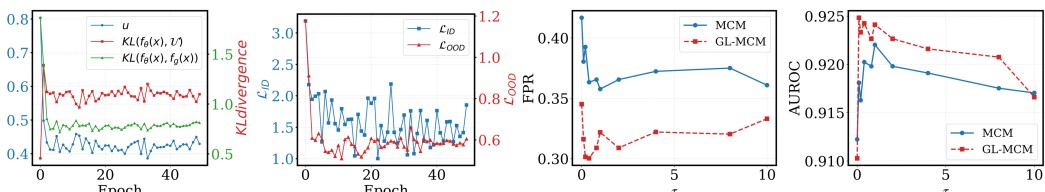

Figure 3: Visualizations of the local uncertainty, the training loss, and OOD performance for different $\tau$ values collectively serve to illustrate the convergence properties of our method.

ratio weakens the KL-based relevance signal, providing insufficient guidance for the optimization process and thus limiting the ability to converge toward an effective soft label assignment. The more result can see the Appendix G for the changes in $u$ under different $\tau$ settings, which matches our analysis.

### 5.4 DISCUSSIONS

**The Convergence and Relevance of Local Uncertainty.** We validate and discuss the convergence and correlation of local uncertainty. As shown in Fig. 3, both divergence measures, $\mathrm{KL}\big(f_g(\mathbf{x}), f_\theta^t(\mathbf{x})\big)$ and $\mathrm{KL}\big(\mathcal{U}, f_\theta^t(\mathbf{x})\big)$, eventually stabilize during model training. Correspondingly, the local uncertainty $u$ converges, consistent with our theoretical expectation that iterative convergence of local uncertainty leads to optimal soft label assignment. Furthermore, the local uncertainty trend exhibits correlations predicted by our method. Specifically, it shows a negative correlation with $\mathrm{KL}\big(f_g(\mathbf{x}), f_\theta^t(\mathbf{x})\big)$ and a positive correlation with $\mathrm{KL}\big(\mathcal{U}, f_\theta^t(\mathbf{x})\big)$. ID and OOD losses, as depicted in the figure, converge simultaneously with the stabilization of $u$. Each resulting OOD loss represents the optimal value achievable by the current model for a given uncertainty level $u$.

**The Dicussion of the Constant Term C.** We discussed the results of the term C independent of u in the generalization error bound. The specific definition of C is provided in the Appendix A. These are two KL divergence results on the OOD test set. Although our loss function does not constrain these two terms, both KL divergences still converge on the training set, as shown in the figure. Additionally, the Appendix I presents KL divergence results for this term on both the training and test sets. These results align closely with our convergence findings on the training set, consistent with many exploratory studies in this area. Furthermore, from the perspective of Rademacher complexity theory, the convergence of these two KL divergence metrics signifies the convergence of empirical error results. The C-term also converges to a constant dominated by empirical error. We present the results in the Appendix I.

**The limitations of smooth labeling.** The feature extraction capability of our method at the patch level is contingent upon the representational power of $f_g$. However, requirements for feature extraction may vary across different real-world datasets. Since our approach relies on the pre-existing knowledge embedded in $f_g$, it may be unable to explore semantic cues beyond the scope of what $f_g$ already captures.

## 6 CONCLUSION

In this paper, we presents a Local Uncertainty Smoothing (LUS) framework to address the challenges in few-shot out-of-distribution detection. Our approach introduces two key innovations, a novel soft label construction method that combines label smoothing with local uncertainty measurement, and a theoretically grounded patch-wise uncertainty mechanism derived from generalization error bound analysis. The framework effectively bridges the distribution gap between in-domain and out-of-domain samples while preserving fine-grained semantic information. Extensive experiments on multiple benchmark datasets demonstrate that our method achieves state-of-the-art performance in few-shot OOD detection scenarios. The results validate both the theoretical foundations and practical effectiveness of our approach. The proposed local uncertainty smoothing strategy provides a robust solution for handling the distribution shift between simulated and real OOD instances.

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

## APPENDIX A  PROOF OF THEOREM 1

First, we define our id label is the y, which means the one-hot distribution for groudtruth and our ood label is the $\mathcal{U}$, which is the uniform distribution. Like the related work, we follow the setup of them and define our id label smooth are as follow:

$$\widetilde{y} = u \cdot f_g(x) + (1 - u) \cdot \mathcal{U} \tag{11}$$

Consider $\ell$ to be the convex logistic loss function applied to binary classification tasks. And considering the property of convex function, we have:

$$\ell\left(f_\theta(x), \widetilde{y}\right) = \ell\left(f_\theta(x), u \cdot f_g(x) + (1 - u) \cdot \mathcal{U}\right) \le (1 - u) \cdot \ell\left(\mathcal{U}, f_\theta(x)\right) + u \cdot \ell\left(f_g(x), f_\theta(x)\right) \tag{12}$$

According to our definition of generalisation error, we have the following:

$$GE(f, \widetilde{y}) = \mathbb{E}_{(x,y)\sim D_{OOD}}\ell\left(f_\theta(x), \widetilde{y}\right)$$

$$= \mathbb{E}_{(x,y)\sim D_{OOD}}\ell\left(f_\theta(x), (1 - u) \cdot \mathcal{U} + u \cdot f_g(x)\right)$$

$$\le \mathbb{E}_{(x,y)\sim D_{OOD}}\left[(1 - u) \cdot \ell(\mathcal{U}, f_\theta(x)) + u \cdot \ell(f_g(x), f_\theta(x))\right]$$

$$= \mathbb{E}_{(x,y)\sim D_{OOD}}\left[\ell(\mathcal{U}, f_\theta(x))\right] - \mathbb{E}_{(x,y)\sim D_{OOD}}\left[u \cdot \ell\left(\mathcal{U}, f_\theta(x)\right)\right] + \mathbb{E}_{(x,y)\sim D_{OOD}}\left[u \cdot \ell(f_g(x), f_\theta(x))\right]$$

$$= \left(1 - \mathbb{E}_{(x,y)\sim D_{OOD}}[u]\right) \cdot \mathbb{E}_{(x,y)\sim D_{OOD}}\left[\ell(\mathcal{U}, f_\theta(x))\right] - Cov\left(u, \ell(\mathcal{U}, f_\theta(x))\right)$$

$$+ \mathbb{E}_{(x,y)\sim D_{OOD}}[u] \cdot \mathbb{E}_{(x,y)\sim D_{OOD}}\left[\ell(f_g(x), f_\theta(x))\right] + Cov\left(u, \ell(f_g(x), f_\theta(x))\right)$$

$$\le Cov\left[u, \ell(f_g(x), f_\theta(x))\right] - Cov\left[u, \ell(\mathcal{U}, f_\theta(x))\right]$$

$$+ \underbrace{\mathbb{E}_{(x,y)\sim D_{OOD}}\left[\ell(\mathcal{U}, f_\theta(x))\right] + \mathbb{E}_{(x,y)\sim D_{OOD}}\left[\ell(f_g(x), f_\theta(x))\right]}_{constant}$$

$$= Cov\left[u, \ell(f_g(x), f_\theta(x))\right] - Cov\left[u, \ell(\mathcal{U}, f_\theta(x))\right] + C \tag{13}$$

Among them, the last two items are defined as irrelevant items $C$ that are irrelevant to $u$. In addition, in many research works (Yuan et al., 2020), the relationship between soft labels and distillation learning is explored. It is believed that by using soft labels and, the loss corresponding to distillation learning can be reduced, that is, $C$ converges to an empirical error, which can also be considered a constant.

Within our theoretical setup and under the stated assumptions, reducing the generalization error bound requires satisfying the conditions that $Cov[u^*, KL(f_g(x), f_\theta(x))] < 0$ and $Cov[u^*, KL(\mathcal{U}, f_\theta(x))] > 0$. This leads to two corollaries for the design of u that it must be negatively correlated with the $KL(f_g(x), f_\theta(x))$, and positively correlated with the $KL(\mathcal{U}, f_\theta(x))$.

## APPENDIX B    OOD SCORE

The CLIP model's multimodal feature alignment capability enables the MCM Ming et al. (2022) method to perform zero-shot OOD detection by quantifying the similarity distribution between image features and $C$ class text embeddings. The OOD Score function is defined as follows:

$$S_{MCM} = \max_i \frac{\exp\left(\langle \phi_I(\mathbf{x}), \phi_T(\mathbf{t}_i) \rangle / \tau\right)}{\sum_{j=1}^C \exp\left(\langle \phi_I(\mathbf{x}), \phi_T(\mathbf{t}_j) \rangle / \tau\right)} \tag{14}$$

where $\tau = 1$ is the temperature parameter, and $\langle \cdot, \cdot \rangle$ denotes cosine similarity.

By introducing a global-local hierarchical feature matching mechanism, GL-MCM Miyai et al. (2025) extends the OOD score calculation to:

$$S_{GL-MCM} = \max_i \frac{\exp\left(\langle \phi_I(\mathbf{x}^{local}), \phi_T(\mathbf{t}_i) \rangle / \tau\right)}{\sum_{j=1}^C \exp\left(\langle \phi_I(\mathbf{x}^{local}), \phi_T(\mathbf{t}_i) \rangle / \tau\right)} + S_{MCM} \tag{15}$$

where $\mathbf{x}^{local}$ represents the feature of the $i$-th local image patch.

## APPENDIX C    EXPERIMENTAL DETAILS

**Base OOD Benchbark**. The implementation of the system adheres to the LoCoOp framework with CLIP-ViT-B/16 Dosovitskiy et al. (2020), where the feature maps exhibit a spatial resolution of 14x14. The key hyperparameters have been empirically configured as follows: the neighbourhood size K = 200 across all experiments, the knowledge distillation coefficient $\alpha = 0.25$, and the regularization weight $\lambda = 0.3$. The additional training specifications encompass 50 epochs with a base learning rate of 0.002, a batch size of 32, and a prompt token length of N=16. It is imperative that all experiments are conducted on a single NVIDIA A6000 GPU in order to ensure hardware consistency.

**Hard OOD Benchbark**. It is evident that our fundamental experimental details are consistent with those of the baseood benchmark. However, given that imagenet-10 and imagenet-20 contain 10 and 20 data types respectively, it was determined that the neighborhood size K=2 would be employed for these hard-to-imitate experiments. The results of the model under the 16-shot setting are presented in full in our paper.

**OpenOOD OOD Benchbark**. The experimental details are fundamentally analogous to the base food benchmark. The imagenet1k has been selected as the ID dataset, while the SSh-hard, NINCO and OpenImage-O have been designated as the OOD dataset. It should be noted that iNaturalist and Texture have not been included in the evaluation process, as these two datasets have previously been evaluated in the base OOD benchmark.

## APPENDIX D    THE SELECTION OF A SUITABLE GENERAL KNOWLEDGE MODEL

Table 6: The cross-domain generalisation performance of prompt-tuned general knowledge models $f_g$, pre-trained on ImageNet-21K and evaluated through out-of-distribution benchmarks.

| Method | OOD Dataset | | | | | | | | | |
|---|---|---|---|---|---|---|---|---|---|---|
| | iNaturalist | | SUN | | Places | | Texture | | Average | |
| | FPR95 | AUROC | FPR95 | AUROC | FPR95 | AUROC | FPR95 | AUROC | FPR95 | AUROC |
| | MCM | | | | | | | | | |
| LUS$_{\text{CLIP}}$ | **27.74** | 94.16 | 34.78 | 93.01 | 42.55 | 90.19 | 48.48 | 89.05 | 38.39 | 91.60 |
| LUS$_{\text{POMP}}$ | 30.80 | **94.17** | **31.25** | **93.91** | **39.78** | **90.79** | **41.50** | **90.81** | **35.83** | **92.42** |
| | GL-MCM | | | | | | | | | |
| LUS$_{\text{CLIP}}$ | **13.59** | **96.81** | 27.73 | 93.87 | 35.94 | 91.09 | 51.21 | 85.80 | 32.12 | 91.89 |
| LUS$_{\text{POMP}}$ | 16.41 | 96.48 | **22.78** | **95.05** | **32.41** | **91.80** | **44.11** | **88.95** | **28.92** | **93.07** |

The following experiments are presented, in which other models of general knowledge are selected to guide the model in acquiring general knowledge. The POMP paper Ren et al. (2023) was selected as

the secondary general knowledge model to present the experimental results. POMP presented the results of prompt tuning on the ImageNet-21K dataset. In this instance, the model under discussion was employed. It is evident that the parameter settings are consistent with the base OOD benchmark. Our results are shown in Table 6, where the clip subscript represents our general knowledge as " a photo of ", and the POMP subscript represents this general knowledge after training on Imagenet-21k. Our results demonstrate that different $f_g(x)$ models can exhibit varying performance for our method, indicating that our model will acquire distinct general knowledge under distinct $f_g(x)$ settings.

Moreover, in order to demonstrate the rationality of our methodology, we employ the same comparison strategy as outlined in Table 1. The results of the ood score of POMP using MCM and GL-MCM in ood detection are presented, as well as the results of the ood score of the LoCoOp model using only our training loss. The following presentation will outline the output results of the model under the KDE strategy. The results of the study are presented in tabular form. The findings of this study suggest that the proposed methodology explores the upper limit of OOD detection, while exhibiting the POMP generalization.

Table 7: The model performance of POMP when used as the $f_g$ model. The present method has been developed in such a manner that it inherits the generalisation ability of POMP, whilst also exploring the upper limit of OOD detection.

| Method | iNaturalist | | SUN | | OOD Dataset
Places | | Texture | | Average | |
|---|---|---|---|---|---|---|---|---|---|---|
| | FPR95 | AUROC | FPR95 | AUROC | FPR95 | AUROC | FPR95 | AUROC | FPR95 | AUROC |
| MCM | | | | | | | | | | |
| LoCoOp | 38.96 | 92.34 | 32.40 | 93.60 | 37.95 | **91.00** | 49.32 | 88.70 | 39.65 | 91.41 |
| LUS | **30.80** | **94.17** | **31.25** | **93.91** | 39.78 | 90.79 | 41.50 | **90.81** | **35.83** | **92.42** |
| GL-MCM | | | | | | | | | | |
| LoCoOp | 24.38 | 94.95 | 25.45 | 94.77 | 32.63 | **91.81** | 52.32 | 86.58 | 33.69 | 92.03 |
| LUS | **16.41** | 96.48 | **22.78** | **95.05** | **32.41** | 91.80 | **44.11** | **88.95** | **28.92** | **93.07** |

# APPENDIX E    MORE EXPERIMENTAL RESULTS

The appendices to this section contain further experimental results of our model, the purpose of which is to demonstrate its experimental performance. The following presentation comprises the experimental results of MCM and GL-MCM under a variety of conditions.

Table 8: cross-domain OOD detection performance comparison across OOD datasets which under different detection frameworks setting: evaluations follow the OpenOOD benchmark with ImageNet-1K as ID data against SSB-hard, NINCO, and OpenImage-O OOD splits, and the MCM cross-evaluation protocol adopting ImageNet-10 ImageNet-20 as ID datasets with reciprocal OOD testing . Our first row represents the id dataset and the second row represents the ood dataset.

| Method | ImageNet-10 | | ImageNet-20 | | ImageNet-1K | | | | | | Average | |
| | ImageNet-20 | | ImageNet-10 | | SSh-hard | | NINCO | | OpenImage-O | | | |
| | FPR95 | AUROC | FPR95 | AUROC | FPR95 | AUROC | FPR95 | AUROC | FPR95 | AUROC | FPR95 | AUROC |
|---|---|---|---|---|---|---|---|---|---|---|---|---|
| LoCoOp | 28.20 | 92.75 | 34.40 | 92.34 | 90.27 | 63.16 | 82.54 | 69.19 | 45.12 | 90.73 | 56.11 | 81.63 |
| Ours | 5.70 | 98.60 | 16.10 | 97.66 | 88.78 | 64.41 | 79.19 | 74.10 | 41.43 | 91.84 | 46.24 | 85.32 |

The experimental results obtained under the OpenOOD and MCM benchmarks demonstrate that GL-MCM exhibits superior performance in cross-dataset ID and OOD detection scenarios when compared to the baseline.

The experimental findings yielded from the execution of MCM benchmarks demonstrate that GL-MCM evinces superior performance in OOD detection scenarios when contrasted with the baseline MCM. This outcome is congruent with our experimental expectations and concomitantly signifies that GL-MCM also attains comparatively favourable enhancement results for GL-MCM of our soft label.

Table 9: OOD detection performance for ImageNet-1k as ID, the SSh-hard, NINCO, OpenImage-O as OOD dataset.

| Method | ImageNet-1K | | | | | |
| | SSh-hard | | NINCO | | OpenImage-O | |
| | FPR95 | AUROC | FPR95 | AUROC | FPR95 | AUROC |
| $\text{LUS}_{\text{MCM}}$ | 88.78 | 64.41 | 79.19 | 74.10 | 41.43 | 91.84 |
| $\text{LUS}_{\text{GL}}$ | 85.13 | 68.27 | 72.57 | 76.06 | 34.59 | 92.36 |

Table 10: OOD detection performance for ImageNet-10, ImageNet-20 as ID, the corresponding imagenet20, imagenet10 as ood datasetas.

| Method | ImageNet10 ImageNet20 | | ImageNet20 ImageNet10 | |
| | FPR95 | AUROC | FPR95 | AUROC |
| $\text{LUS}_{\text{MCM}}$ | 5.70 | 98.60 | 16.10 | 97.66 |
| $\text{LUS}_{\text{GL}}$ | 10.60 | 98.66 | 9.90 | 98.32 |

The subsequent presentation will expound upon the findings of the model's image detection process in relation to imaget100, which will be utilised as the ID data. The experimental results of the model on 4-shot are also presented. In the present experiment, the value of K was set to 20. The 1-shot configuration was not selected as the experimental outcome due to the inability of our model to converge on the original LoCoOp setting. In order to conduct a one-shot experiment, it is necessary to enlarge the epoch under the LoCoOp setting until the experimental results obtained are consistent with those reported in the aforementioned paper. The present study employs imagenet-100 as the ID dataset, thereby adopting a methodology that explores enhanced object detection while ensuring the efficacy of the $f_g(x)$ model. This approach is employed to demonstrate the efficacy of the proposed methodology.

## APPENDIX F  COMPARING WITH MORE UNCERTAINTY METHOD.

**Static weight.** We first define the static method which use the weight is 1/2. We define the soft label for OOD data as follows:

$$\widetilde{y} = \frac{1}{2} \cdot f_g(x) + \frac{1}{2} \cdot \mathcal{U} \tag{16}$$

**Max logit.** We initially define the uncertainty measure as the maximum logit, denoted as:

$$u = \max_{c \in \mathcal{C}} f_c(\mathbf{x}) \tag{17}$$

where $f_c(\mathbf{x})$ is the logit output for class $c$ given input $\mathbf{x}$, and $\mathcal{C}$ is the set of all classes.

Since this raw uncertainty value is not normalized, we scale it to the range $[0, 1]$ using extremal statistics from the entire training dataset $\mathcal{D}_{\text{train}}$. Let:

$$u_{\min} = \min_{\mathbf{x}_i \in \mathcal{D}_{\text{train}}} \max_c f_c(\mathbf{x}_i) \tag{18}$$

$$u_{\max} = \max_{\mathbf{x}_i \in \mathcal{D}_{\text{train}}} \max_c f_c(\mathbf{x}_i) \tag{19}$$

represent the global minimum and maximum uncertainty values observed over $\mathcal{D}_{\text{train}}$. The normalized uncertainty $u_{\text{norm}}$ is then defined as:

$$u_{\text{norm}} = \frac{u - u_{\min}}{u_{\max} - u_{\min}} \tag{20}$$

This min-max normalization ensures $u_{\text{norm}} \in [0, 1]$ with the property that the most uncertain sample in the training set maps to 1 and the least uncertain to 0.

$$u = \frac{u - u_{min}}{u_{max} - u_{min}} \tag{21}$$

Table 11: OOD detection performance for ImageNet-10, ImageNet-20 as ID, the corresponding imagenet20, imagenet10 as ood datasetas.

| Method | ImageNet10 ImageNet20 | | ImageNet20 ImageNet10 | |
|---|---|---|---|---|
| | FPR95 | AUROC | FPR95 | AUROC |
| $\text{LUS}_{\text{MCM}}$ | 5.70 | 98.60 | 16.10 | 97.66 |
| $\text{LUS}_{\text{GL}}$ | 10.60 | 98.66 | 9.90 | 98.32 |

Table 12: Cross-domain generalization performance on ImageNet-100 as ID data under four-shot learning protocol. A comparison was made between MCM and LoCoOp.

| Method | iNaturalist | | SUN | | OOD Dataset Places | | Texture | | Average | |
|---|---|---|---|---|---|---|---|---|---|---|
| | FPR95 | AUROC | FPR95 | AUROC | FPR95 | AUROC | FPR95 | AUROC | FPR95 | AUROC |
| | | | | | MCM | | | | | |
| $\text{LoCoOp}_{\text{MCM}}$ | 18.69 | 96.54 | 21.16 | 96.32 | 27.82 | 95.12 | 26.17 | 94.99 | 23.46 | 95.74 |
| $\text{LUS}_{\text{MCM}}$ | **10.70** | **97.71** | **16.81** | **96.92** | **22.52** | **95.65** | **24.68** | **95.49** | **18.67** | **96.44** |
| | | | | | GL-MCM | | | | | |
| $\text{LoCoOp}_{\text{GL}}$ | 12.97 | 97.09 | **12.55** | 97.20 | **18.15** | 96.06 | **26.17** | 94.36 | 17.46 | 96.18 |
| $\text{LUS}_{\text{GL}}$ | 4.44 | 98.87 | 13.15 | **97.42** | 18.43 | 96.11 | 27.23 | **94.48** | **15.81** | **96.72** |

**Entropy.** The entropy-based uncertainty is defined as $u = -\sum_c p_c(\mathbf{x}) \log p_c(\mathbf{x})$ and normalized to [0,1] using:

$$u_{\text{norm}} = \frac{u - u_{\min}}{u_{\max} - u_{\min}} \tag{22}$$

where $u_{\min}$ and $u_{\max}$ are the extreme entropy values from the training set.

## APPENDIX G  MORE TEMPERATURE COEFFICIENT VISUALIZATION RESULTS.

This section analyzes the convergence of u under different hyperparameter settings in the paper. These images match our analysis in the article. For smaller temperature coefficients, $u$ will have large fluctuations, while for larger temperature coefficients, the fluctuations are smaller, but the performance deteriorates. In the experiments in the paper, we choose the results when the temperature coefficient is 1.

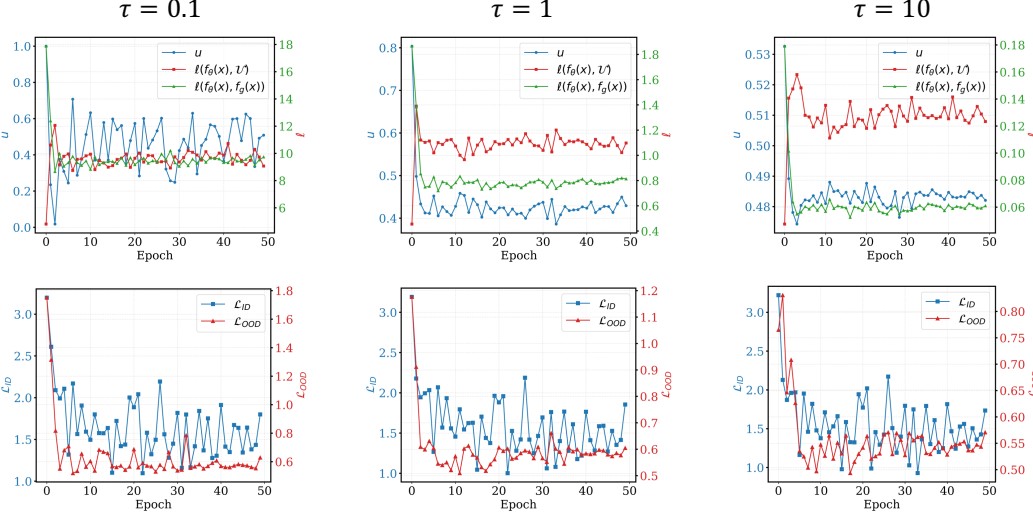

Figure 4: More hyperparameter $\tau$ visualization results.

## APPENDIX H  THE EXPERIMENT ON GEB COMPONENT CONSTANT.

This part shows the results of $C$ on the test set. Obviously, for LoCoOp, the KL divergence for general knowledge is large. For our method, both KL divergences on the test set remain small, which explains one of the reasons why we consider these two terms as constant terms in the generalization error

Table 13: Two KL divergence results on the test set.

| Method | $KL(f_\theta(x), \mathcal{U})$ | $KL(f_\theta(x), f_g(x))$ |
|---|---|---|
| LoCoOp | 0.84 | 1.89 |
| Ours | 0.81 | 1.13 |

## APPENDIX I  OOD DATASETS.

**iNaturalist.** The dataset under consideration is comprised of 859,000 biological specimens, which are divided into more than 5,000 taxonomic categories. The primary focus of the dataset is flora and fauna biodiversity. In accordance with the established protocol, the evaluation process is conducted using a sample of 10,000 images, selected at random from a total of 110 classes, with the exclusion of those that are already present in the ImageNet-1K database.

**SUN.** The scene recognition corpus under consideration contains 130,000 visual instances, which are divided into 397 environmental categories. For the purpose of comparative analysis, a curated subset of 10,000 images has been employed, sampled from 50 ImageNet-disjoint classes.

**Places.** Places provides complementary coverage of environmental semantics, mirroring SUN's conceptual scope in scene understanding. The assessment utilises 10,000 images from 50 non-overlapping classes.

**TEXTURE.** The present corpus is one that has been specifically compiled for the purpose of this study. It consists of 5,640 high-resolution texture patterns that have been organised into 47 material categories. A comprehensive evaluation is performed using the full dataset.

**OpenImage-O.** This rigorously curated visual recognition benchmark comprises 17,632 images that have been manually filtered through multi-stage quality assurance protocols, achieving 7.8× greater scale diversity than ImageNet-O through pixel-coverage optimisation.

**SSB-hard.** Derived from ImageNet-21K's hierarchical ontology through semantic scarcity sampling, this 49,000-image benchmark spans 980 visually complex categories characterised by high inter-class ambiguity.

**NINCO.** The dataset contains 5,879 meticulously annotated samples across 64 novel categories, thereby introducing conceptual novelty through systematic exclusion of ImageNet-1K semantic overlaps.

**ImageNet-10.** The creation of ImageNet-10 was driven by the necessity to emulate the class distribution of CIFAR-10, while incorporating high-resolution images. The following categories are contained within the dataset, along with their respective class identifiers: The following subject headings have been identified: The following terms are listed: 'warplane' (n04552348), 'sports car' (n04285008), 'brambling bird' (n01530575), 'Siamese cat' (n02123597), 'antelope' (n02422699). The following have been identified: 'Swiss mountain dog' (n02107574), 'bull frog' (n01641577), 'garbage truck' (n03417042), 'horse' (n02389026), and 'container ship' (n03095699).

**ImageNet-20.** In order to facilitate the evaluation of hard OODs with realistic datasets, ImageNet-20 has been curated. The dataset under consideration consists of 20 classes that are semantically similar to ImageNet-10. The categories are selected based on the distance in the WordNet synsets. The following categories are contained therein: The following items are listed herewith: The following objects are documented: a sailboat (n04147183), a canoe (n02951358), a balloon (n02782093), a tank (n04389033), a missile (n03773504), and a bullet train (n02917067). The following species were documented: A starfish (n02317335), a spotted salamander (n01632458), a common newt (n01630670), a zebra (n01631663), and a frilled lizard (n02391049). For the purposes of this study,

the following taxa were selected: the green lizard (n01693334), the African crocodile (n01697457), the Arctic fox (n02120079), the timber wolf (n02114367), the brown bear (n02132136), the moped (n03785016), the steam locomotive (n04310018), the space shuttle (n04266014) and the snowmobile (n04252077).

