# OpenReview forum: "Local Uncertainty Smoothing For Few-shot OOD Detection"
_ICLR.cc/2026/Conference — ICLR 2026 Conference Withdrawn Submission_

### Official Review · Reviewer_PHB6 · 2025-10-26

**Soundness:** 3
**Presentation:** 2
**Contribution:** 2
**Rating:** 2
**Confidence:** 5

**Summary:**

The authors propose a novel framework addressing the problem of few-shot out-of-distribution (OOD) detection. They tackle the limitations of previous methods that relied on pseudo-OOD labels with uniform distribution by introducing label smoothing and local uncertainty techniques. The paper provides a theoretical proof for the Generalized Energy Bound (GEB) and conducts experimental validation on the task.

**Strengths:**

1. The paper is well-written and easy to follow.

2. It correctly identifies a key limitation in existing methods that use a uniform distribution for pseudo-OOD labels.

3. The proposed solution of using label smoothing and local uncertainty effectively addresses this issue and leads to a measurable performance improvement.

**Weaknesses:**

1. The ablation study lacks removing uniform pseudo-OOD distribution in EQ.4

2. The novelty and scope of the proposed framework is constrained. The experimental comparisons are not fully conclusive. The benchmarked methods are not state-of-the-art, and there is a notable absence of comparisons with recent, strong contenders such as SCT (NeurIPS 2024) and Local-Prompt (ICLR 2025). Furthermore, the reported performance gains, while present, are not overwhelming against the older baselines, raising questions about the method's competitiveness.

3. The design and purpose of the fg in EQ4 are ambiguous and require further elaboration. The description suggests it involves re-processing low-similarity patches through a model pre-trained on ImageNet-21k, using ID labels. Several critical questions arise:
- What is the underlying intuition? The authors should clarify the theoretical or empirical motivation behind this specific operation. Is it intended as a form of knowledge distillation? If so, the connection should be explicitly stated and justified.
- How can it provide meaningful guidance? Since the label space of ImageNet-21k is largely unrelated to the ood patches, the logits produced by fg may not offer semantically relevant guidance for the target domain.

4. There is a minor typo in line 252 of the paper: "uis" is missing a space.

**Questions:**

See weaknesses.

---

### Official Review · Reviewer_6xLL · 2025-10-27

**Soundness:** 2
**Presentation:** 1
**Contribution:** 2
**Rating:** 2
**Confidence:** 3

**Summary:**

The authors propose an adaptive label-smoothing method to increase prompt-learning-based OOD detection with VLMs. The strength of the smoothing depends on the ratio between the KL divergence of the predicted distribution to the uniform distribution and to a general knowledge model. In this way, patches that are unrelated to the ground truth label can be used as stronger OOD samples than more relevant patches. The method is evaluated against other prompt learning baselines on a range of standard datasets.

**Strengths:**

- The idea to adapt the label strength of OOD patches is clear and makes sense.
- Figure 1 clearly illustrates the core idea behind the proposed method.
- Various aspects of the proposed system are ablated.

**Weaknesses:**

The presentation of the paper has several issues. First, LLM-like output not only appears in many sections but is also completely unrelated in some cases. For instance, the text suddenly talks about how the proposed OOD detection method "enhances object detection performance" (L412) or adopts "a methodology that explores enhanced object detection" (L780). Another example is on L712-713, which suddenly mentions a KDE strategy for the first time: "The following presentation will outline the output results of the model under the KDE strategy. The results of the study are presented in tabular form."
Second, the clarity of the method section should be improved. For instance, f_g is never explained beyond being "a general knowledge model that denotes the semantic understanding of patch-level OOD data", the distributions used as input to the KL divergences in Figure 2 are different from the text, and notation contains some issues in cases such as Eq. 3, where it should be specified that x is sampled from the ID and OOD distributions, or the repetition of KL(U, fθ(x)) in Theorem 1.
Third, there are still many typos/mistakes in the text. Some examples are "This offers a new understand" (L89), "we propose local uncertainty smoothing method which refining the label distribution" (L215), "value of uis critical" and "between uand the model" (L252-254), "And and" (L266), "The more result can see the Appendix G" (L444) and so on.
Overall, these issues combined make the manuscript and its proposed contributions far less clear.

The related work and benchmark also miss some more recent works that improve upon LoCoOp, such as [1, 2], which should be included in the comparison. The numbers reported for methods such as NegPrompt are also (far) lower than those reported in their original paper, where they outperform the proposed method, which should be explained.

[1] Lafon, Marc, et al. "Gallop: Learning global and local prompts for vision-language models." European Conference on Computer Vision. Cham: Springer Nature Switzerland, 2024.
[2] Zhang, Yabin, and Lei Zhang. "Adaneg: Adaptive negative proxy guided ood detection with vision-language models." Advances in Neural Information Processing Systems 37 (2024): 38744-38768.

**Questions:**

- Why is the comparison in Table 3 only done w.r.t. LoCoOp, which version is it, and which version of LUS is used? How come the gap in performance between the proposed method and LoCoOp is much larger than on other Near-OOD cases, such as ImageNet:SSB-Hard in Table 8?
- The motivation of the method is not justified appropriately. Why is the uncertainty of the model being trained a good signal for how strongly a patch should be considered as OOD?
- The method section should make clear what LUS_MCM and LUS_GL are.
- It is unclear what exactly is ablated in the ablation analysis of Table 5. Moreover, f_g does not seem to have a real benefit.
- The best performing models should be bolded for Texture in Table 1.
- How sensitive is the proposed method to the chosen hyperparameters?
- It is unclear what the additional computational cost is for training compared to LoCoOp.

---

### Official Review · Reviewer_vNYK · 2025-10-31

**Soundness:** 2
**Presentation:** 1
**Contribution:** 1
**Rating:** 2
**Confidence:** 5

**Summary:**

This paper investigates the problem of assigning uniform labels to extracted local OOD features in existing few-shot OOD detection methods. It proposes a Local Uncertainty Smoothing (LUS) framework that adaptively determines optimal soft label assignments for each local OOD patch. Experiments on the ImageNet-1k OOD benchmark are conducted to evaluate the proposed method.

**Strengths:**

1. It’s an interesting research problem that assigning uniform labels to local OOD features can degrade ID classification accuracy.

2. This paper provides a theoretical analysis of the proposed method.

**Weaknesses:**

1. **This paper lacks a direct experimental comparison with two highly relevant and strong baselines, SCT [1] and Local-Prompt [2].** The performances of these two methods, reported in their papers, are significantly better than LUS on ImageNet-1k OOD benchmark.
2. **This paper doesn’t present clear mathematical formulations for some key modules of the proposed framework**, which makes it harder for readers to understand the technical details. To be exact, there are no mathematical formulations for $f_\theta(x)$ and $f_g(x)$ in the main text and the term “general knowledge model” is not clearly defined. In addition, the meaning of $\omega$ in Equation (6) is not explained.
3. **There are many typing errors, grammar errors or confusing expressions in the main text,** which makes it harder for readers to understand the proposed framework. For instance: “while enhance” in Line 81, “a new understand of the ood local features” in Line 89, “one-shot distributions” in Line 214, “which refining” in Line 215, Line 231 says “$u$ denotes an uncertainty method”, “uand” in Line 254, and so on.
4. The reviewer suggests that the authors **add experiments on different CLIP backbone architectures and more hard OOD detection tasks,** as in [1], to provide a more comprehensive evaluation of the proposed method.

[1] Geng Yu, et al. Self-Calibrated Tuning of Vision-Language Models for Out-of-Distribution Detection.

[2] Fanhu Zeng, et al. Local-Prompt: Extensible Local Prompts for Few-Shot Out-of-Distribution Detection.

**Questions:**

Please see the weaknesses.

**Details Of Ethics Concerns:**

The reviewer doesn't notice any ethical issues with this paper.

---

### Official Review · Reviewer_aoub · 2025-10-31

**Soundness:** 2
**Presentation:** 2
**Contribution:** 2
**Rating:** 4
**Confidence:** 4

**Summary:**

The paper proposes a novel Local Uncertainty Smoothing (LUS) framework for few-shot out-of-distribution (OOD) detection. Current approaches often rely on auxiliary OOD data derived from in-distribution (ID) samples (e.g., local image patches), but such artificially constructed OOD instances can differ significantly from real OOD data, leading to unstable learning. LUS addresses this by integrating label smoothing and a patch-wise local uncertainty measure. Theoretically, the paper analyzes the relationship between local uncertainty and KL divergence from a generalization error bound (GEB) perspective, providing a foundation for dynamic soft label assignment. Experimentally, the method is validated on benchmarks like ImageNet-1K, iNaturalist, and hard OOD datasets, showing state-of-the-art performance in few-shot settings.

**Strengths:**

1. The LUS framework introduces a fresh perspective on OOD detection by moving beyond hard labels (e.g., uniform distributions) to soft labels based on general knowledge and uncertainty.
2. The paper provides a solid theoretical foundation through Theorem 1, which connects local uncertainty to KL divergence via a generalization error bound.
3.  The experimental evaluation is thorough, covering multiple OOD benchmarks and metrics (FPR95, AUROC, ID accuracy).

**Weaknesses:**

1. The paper identifies the temperature coefficient $\tau$ as critical for balancing KL divergence terms. While $\tau$=1 is optimal in experiments, tuning it for new datasets may be non-trivial and computationally expensive, potentially hindering plug-and-play use.
2.  Experiments focus on image data and near-OOD scenarios, lacking cross-domain tests (e.g., text or video). This reduces insights into generalizability.
3. This paper's foundational motivation aligns with prior research in OOD detection. A notable example is the Energy-based method, which demonstrates the inadequacy of uniform distributions and advances an energy-bound solution [1] to optimize detection performance.
4. The paper does not discuss efficiency trade-offs, which could be critical for resource-constrained environments.

[1] Liu, Weitang, et al. "Energy-based out-of-distribution detection." Advances in neural information processing systems 33 (2020): 21464-21475.

**Questions:**

1. How does LUS perform on "far-OOD" data (e.g., adversarial OOD examples)?
2. Is the method adaptable to non-visual domains (e.g., NLP, video) ?
3. How about the efficiency of the proposed method?

---

### Note · Authors · 2026-01-10

I have read and agree with the venue's withdrawal policy on behalf of myself and my co-authors.